# Differential Impacts on Tensional Homeostasis of Gastric Cancer Cells Due to Distinct Domain Variants of E-Cadherin

**DOI:** 10.3390/cancers14112690

**Published:** 2022-05-29

**Authors:** Han Xu, Katie A. Bunde, Joana Figueiredo, Raquel Seruca, Michael L. Smith, Dimitrije Stamenović

**Affiliations:** 1Department of Biomedical Engineering, Boston University, Boston, MA 02215, USA; hanxu07@bu.edu (H.X.); kbunde@bu.edu (K.A.B.); 2Epithelial Interactions in Cancer, Instituto de Investigação e Inovação em Saúde (i3S), University of Porto, 4200-135 Porto, Portugal; jfigueiredo@ipatimup.pt (J.F.); rseruca@ipatimup.pt (R.S.); 3Institute of Molecular Pathology and Immunology of the University of Porto (IPATIMUP), 4200-135 Porto, Portugal; 4Medical Faculty, University of Porto, 4200-319 Porto, Portugal; 5Division of Material Science and Engineering, Boston University, Brookline, MA 02446, USA

**Keywords:** tensional homeostasis, traction microscopy, gastric cancer cells, E-cadherin mutations, extracellular matrix proteins

## Abstract

**Simple Summary:**

Tensional homeostasis describes the ability of cells and tissues to maintain their internal mechanical tension stable at a set point value. A breakdown of tensional homeostasis is the hallmark of disease progression, including cancers. In cancers of epithelial origin, this phenomenon is closely associated with dysfunction of E-cadherin. In this study, we investigated how E-cadherin mutations identified in patients with cancer affect tensional homeostasis. Our results show that mutations affecting the juxtamembrane and intracellular domains of E-cadherin are detrimental for tensional homeostasis of gastric cancer cells.

**Abstract:**

In epithelia, breakdown of tensional homeostasis is closely associated with E-cadherin dysfunction and disruption of tissue function and integrity. In this study, we investigated the effect of E-cadherin mutations affecting distinct protein domains on tensional homeostasis of gastric cancer cells. We used micropattern traction microscopy to measure temporal fluctuations of cellular traction forces in AGS cells transfected with the wild-type E-cadherin or with variants affecting the extracellular, the juxtamembrane, and the intracellular domains of the protein. We focused on the dynamic aspect of tensional homeostasis, namely the ability of cells to maintain a consistent level of tension, with low temporal variability around a set point. Cells were cultured on hydrogels micropatterned with different extracellular matrix (ECM) proteins to test whether the ECM adhesion impacts cell behavior. A combination of Fibronectin and Vitronectin was used as a substrate that promotes the adhesive ability of E-cadherin dysfunctional cells, whereas Collagen VI was used to test an unfavorable ECM condition. Our results showed that mutations affecting distinct E-cadherin domains influenced differently cell tensional homeostasis, and pinpointed the juxtamembrane and intracellular regions of E-cadherin as the key players in this process. Furthermore, Fibronectin and Vitronectin might modulate cancer cell behavior towards tensional homeostasis.

## 1. Introduction

Tensional homeostasis is defined as the ability of cells to maintain their endogenous mechanical tension stable, at a preferred set point value [1,2,3]. A breakdown in tensional homeostasis is the hallmark of several diseases, including cancer [4,5]. In malignant epithelial cells, breakdown of tensional homeostasis is closely associated with E-cadherin dysfunction and disruption of tissue function and integrity [6,7].

E-cadherin is a main adhesion molecule that coordinates a mechanical circuit of cell-cell linkages, contractile forces and biochemical signals to sustain a functional epithelial barrier [8]. The E-cadherin extracellular domain is responsible for the homophilic binding of E-cadherin molecules on neighboring cells, assuring cohesion and force transmission across the epithelia. On the other hand, the intracellular portion of E-cadherin molecules is coupled with the cytoskeleton, increasing cytoskeletal stiffness and stress resistance during cell rearrangements, such as those occurring in cell division [9,10]. Therefore, it is commonly accepted that E-cadherin is a potent tumor suppressor and is involved in limiting tumor cell migration. Accordingly, genetic and epigenetic alterations of E-cadherin are observed in 70% of carcinomas and are associated with invasion and metastasis [11].

Recent studies suggest that along with loss of cell-cell adhesion, cancer cells may undergo an excessive deposition of extracellular matrix (ECM) proteins, such as collagen [12,13], in an attempt to stabilize their cytoskeletal tension through cell-matrix force transmission. This enhancement of matrix deposition creates a scaffold that contributes for cancer development by forming a physical barrier to anticancer drugs, providing growth factor and cytokines reserves, and promoting cell-ECM adhesion for successful invasion and proliferation [4,14,15,16]. However, the impact of E-cadherin dysfunction on mechanical forces dictating abnormal cell-ECM dynamics remains to be unraveled.

During the last decade, it has been demonstrated that tensional homeostasis is tightly regulated by cytoskeletal tension and by traction forces occurring in isolated cells or in cellular clusters [17,18,19,20,21]. Furthermore, we found that clusters of endothelial cells exhibit decreased temporal fluctuations of traction forces, when compared to single cells, suggesting intercellular adhesions as relevant factors for tensional homeostasis and multicellular contexts as favorable mechanical environments [18,20]. However, in gastric cancer cells transfected with the wild-type E-cadherin, clustering did not cause as significant attenuation of temporal fluctuations of traction forces as in the case of endothelial cells [20].

In the present study, we investigated the effect of cancer-associated variants of E-cadherin in intracellular force transmission and in tensional homeostasis of gastric cancer cells, taking into account cell’s interplay with the ECM. For that purpose, gastric cancer cells transfected with the wild-type (WT) E-cadherin or with mutants affecting the extracellular, the juxtamembrane, and the intracellular domains of the protein were assayed in specific ECMs and subsequently, evaluated for traction forces, as well as their temporal variability. In particular, we focused on the dynamic aspect of tensional homeostasis, namely the ability of cells to maintain a consistent level of tension, with a low temporal variability around a set point [3,21]. Our results showed that variants located in distinct protein domains yield different cell mechanic profiles, and pinpointed the juxtamembrane and the intracellular regions of E-cadherin as the key players in this process. Ultimately, our data indicated that ECM components such as Fibronectin and Vitronectin might modulate cancer cell behavior towards tensional homeostasis.

## 2. Materials and Methods

### 2.1. Cell Culture and Transfections

Cells were cultured as previously described [22]. Briefly, AGS cell line (gastric adenocarcinoma, ATCC number CRL-1739) was maintained in RPMI medium (Gibco, Invitrogen) supplemented with 10% fetal bovine serum (HyClone, Perbio) and 1% penicillin/streptomycin (Gibco, Invitrogen). Cells were incubated at 37 °C under 5% CO_2_ humidified air. Transfections were performed using Lipofectamine 2000 (Invitrogen), according to the manufacturer’s recommendations. For transfections, we have used 1 μg of DNA of vectors encoding the WT E-cadherin or the A634V (extracellular domain mutant), R749W (juxtamembrane domain mutant), and V832M (intracellular domain mutant) variants, as well as the empty vector (Mock). These E-cadherin variants are described as causative of hereditary diffuse gastric cancer syndrome (HDGC) [23,24,25,26]. Transfected cells were selected by antibiotic resistance to blasticidin (5 μg/mL; Gibco, Invitrogen). At the end of each transfection, putative cytotoxic effects were evaluated by analysing cell viability.

### 2.2. Micropattern Traction Microscopy

An indirect patterning method was used to create polyacrylamide (PAA) gels with a grid of covalently bound dots of 250 μg/mL AlexaFluor-488 tagged Collagen VI (Col VI, Thermo Fisher, Waltham, MA, USA) or of a protein mix composed by 125 μg/mL AlexaFluor-488 tagged Fibronectin (Fn) and 125 μg/mL Vitronectin (Vt), as described previously [20,22]. The Fn + Vt combination was used to test an ECM substrate that promotes the adhesive ability of E-cadherin dysfunctional cells, whereas the patterning with Col VI was used to test an unfavorable ECM condition [22]. The patterns were made up of 2-μm diameter dots at 6 μm center-to-center separation. The PAA gels had an elastic modulus of *E* ≈ 6.7 kPa and a poisons ratio of ν = 0.445, as determined previously [27,28]. A suspension of 3–5 × 10^4^ cells/mL was seeded on micropatterned gels, which were then incubated for 24 h to allow the establishment of focal adhesions (FAs) at the micropatterned dots. Due to the non-fouling properties of the PAA gel, the formation of FAs is limited to the micropatterned dots. Consequently, traction forces can be modeled as discrete forces applied to the individual dots [27].

Cells were subsequently imaged with an Olympus IX881 microscope and a Hamamatsu Orca R2 camera. Images were taken every 5 min for 1 h (13 images). Experiments were carried out in a chamber under controlled environment and maintaining 37 °C, 70% humidity and 5% CO_2_. Images capturing the cells and the fluorescent dot array were analyzed using custom MATLAB scripts, as reported by Polio et al. [27]. The program determines the displacement vector (**u**) of the geometrical center of each dot and calculates the corresponding tangential traction force vector (**F**) as follows,
(1)$$F=\frac{\pi Eau}{2+\nu -\nu^{2}},$$
where *a* = 1 μm is the radius of the dot markers [29]. The method assumes that displacements of individual dots result primarily from the traction forces applied at a specific dot and not from forces applied at the adjacent positions. This is reasonable considering that the center-to-center distance between the dots is three dot diameters and that the displacements decreases inversely with the dot radius [27].

### 2.3. Contractile Moment and Tension

The magnitude of the contractile moment (*M*) was used as a quantitative metric of the magnitude of the cell traction field [18,20,30]. Physically, *M* > 0 represents a strength by which the contracting cell “pinches” the substrate as it probes its rigidity; *M* < 0 is not physically feasible. The significance of *M* is that, for a plane state of stress in the cell (i.e., two-dimensional state of stress), *M* is equivalent to the mean normal stress (tension) within the cell times the cell volume [31]. Given that cells do not change their volume during the experiments, *M* is a direct indication of the cytoskeletal tension.

The contractile moment was calculated at each 5-min time interval (*t*) as follows,
(2)$$M(t)=\sum_{i=1}^{N}r_{i}(t)\cdot F_{i}(t),$$
where **r***_i_* denotes the position vector of the center of a micropatterned dot (i.e., a moment arm vector), **F***_i_* is the corresponding traction force vector, the dot denotes the scalar product between the vectors, and *N* is the number of FAs within a cell. Of note, *N* is equal to the number of traction forces acting at the micropatterned dots at a given instant.

### 2.4. Data Analysis

For each image taken, measured traction forces were adjusted to satisfy mechanical equilibrium as described previously [31]. If this equilibration process yielded forces of unusually high magnitudes (>15 nN), those cells were excluded from the analysis. Traction forces below 0.3 nN were also not considered since displacements corresponding to these forces were indistinguishable from background noise [21].

For each 5-min time interval, *M*(*t*) was calculated according to Equation (2). Cells where *M*(*t*) < 0 in more than 3 (out of 13) time intervals were discarded from further analysis. Otherwise, negative *M*s were replaced by zero values. For each cell, we computed the time-average value (〈*M*〉) of *M*(*t*) and the corresponding standard deviation (*SD_M_*) over the 1-h observation time. The coefficient of variation (*CV_M_*) was subsequently obtained as the ratio of both parameters, *CV_M_* = *SD_M_*/〈*M*〉 [20].

For each FA identified within a cell, we computed the time-average traction force (〈*F*〉), the corresponding standard deviation (*SD_F_*), and the corresponding coefficient of variation (*CV_F_*) as *CV_F_ = SD_F_*/〈*F*〉. Values of 〈*F*〉 > 0.3 nN obtained from all FAs were then sorted in an ascending manner; the difference between the highest and lowest values of 〈*F*〉 was calculated and divided by ten. Hence, ten bins of data were obtained for 〈*F*〉 and the corresponding *CV_F_*. For each bin, we calculated the mean values of 〈*F*〉 and of *CV_F_* and the corresponding standard errors [21].

### 2.5. Quantitative Metrics of Tensional Homeostasis

The coefficients of variation *CV_M_* and *CV_F_* indicate the extent of temporal variability of *M*(*t*) and *F*(*t*) relative to 〈*M*〉 and 〈*F*〉, respectively. Thus, we used *CV_M_* and *CV_F_* as quantitative metrics of tensional homeostasis at the whole cell level and at the FA level, respectively. As *CV_M_* and *CV_F_* approach zero values, it indicates that a cell and its FAs are close to the state of tensional homeostasis.

### 2.6. Statistical Analysis

For statistical analysis, median values ± median absolute deviation (MAD = the median of the absolute deviations from the data’s median) were compared using the Mann-Whitney Rank Sum Test since data for 〈*M*〉 of individual cells did not exhibit a normal distribution. Normality of the distribution was evaluated through the Shapiro-Wilk test. Significance was established at *p* < 0.05 or *p* < 0.1, as indicated. The statistical analysis was carried out using SigmaPlot software (version 13, Chicago, IL, USA).

## 3. Results

In biophysics, biomechanics and mechanobiology, the contractile moment is used as a quantitative metric of the strength of cell contractility [18,20,30]. The contractile machinery of cells generates traction forces, which cells apply to the substrate via focal adhesions, thus inducing substrate deformation. This substrate contraction, sometimes referred to as “pinching”, is used by cells for mechanosensing substrate rigidity, which is essential for maintenance of tensional homeostasis [5]. Aside from the applied traction forces, the magnitude of this contraction also depends on the cell geometry (given by the moment arm in Equation (2)). For example, if two cells would exert the same traction force vectors on the substrate, a more spread cell would have the greater contractile moment.

### 3.1. Contractile Moments of Juxtamembrane, Intracellular E-Cadherin Mutants, and Mock Cells Exhibit Greater Temporal Variability than Wild-Type Cells and Extracellular Mutants

To investigate the effects of E-cadherin dysfunction in cellular traction forces, we used cells transfected with the WT E-cadherin or variants shown to be causative of hereditary diffuse gastric cancer syndrome (HDGC) [23,24,25,26]. Importantly, we have selected variants affecting different protein domains to evaluate potential domain-specific functions: A634V affects the extracellular domain, R749W affects the juxtamembrane domain, and V832M affects the intracellular portion of the protein (Figure 1).

Traction microscopy measurements of cells were carried out on Fn + Vt micropatterns, which was described as an advantageous substrate for adhesion of E-cadherin mutant cells. For comparison of *M*(*t*) between different cells, we normalized *M*(*t*) with its time average 〈*M*〉. Time lapses of *M*(*t*)/〈*M*〉 exhibited erratic temporal fluctuations over the 1-h observation time in all cell types (Figure 2). However, the dynamics of the WT (Figure 2A) and A634V cells (Figure 2B) was less fluctuating than that of the R749W (Figure 2C), V832M (Figure 2D), and Mock cells (Figure 2E).

### 3.2. E-Cadherin Expression Promotes Cell Tension

The cells transfected with the WT E-cadherin or the different mutants had significantly greater median values of 〈*M*〉 than the Mock cells, which do not express E-cadherin (*p* < 0.001 for WT and A634V; *p* = 0.005 for R749W; *p* = 0.013 for V832M; Figure 3A). This suggests that the presence of E-cadherin promotes cell contractility, regardless of whether it is a mutated variant. Median values of 〈*M*〉 of the mutants were not significantly different from the WT cells. Interestingly, the V832M cytoplasmic mutant presented an evident, although not significant (*p* = 0.151), decrease in 〈*M*〉 when compared with the WT cells, further suggesting its deleterious effect. There was, however, a significant difference between the V832M and A634V cells, where the median 〈*M*〉 of the former was significantly smaller than that of the latter (*p* = 0.034).

To evaluate whether the observed differences in 〈*M*〉 might be explained by differences in the cells’ ability to establish FAs, we computed the median number of FAs in each cellular condition. The Mock cells exhibited a significantly lower median number of FAs (*p* < 0.05) than all the E-cadherin-transfected cells (Figure 3B), which was consistent with the significantly lower median value of 〈*M*〉 of the Mock cells in comparison with all the transfected cells (Figure 3B vs. Figure 3A). The R749W cells exhibited significantly greater number of FAs in comparison with the WT cells (*p* = 0.011), in accordance with the difference between their respective median values of 〈*M*〉, albeit non-significant (Figure 3B vs. Figure 3A). On the other hand, the A634V and V832M cells had nearly the same median number of FAs although their respective median values of 〈*M*〉 were significantly different (Figure 3B vs. Figure 3A). Together, these results suggest that, aside the differences in the number of FAs, other factors may contribute to the differences in 〈*M*〉 observed across cell variants. For instance, the magnitude of traction forces applied at individual FAs, the extent of cell spreading, and the cell geometry (shape, volume) may impact cell mechanical performance.

### 3.3. Juxtamembrane and Intracellular E-Cadherin Mutants Compromise Tensional Homeostasis

We next investigated the temporal variability of the contractile moment in our cell lines using *CV_M_* as a metric of tensional homeostasis. We verified that the R749W and V832M mutants, as well as the Mock cells had significantly higher median values of *CV_M_* when compared with the WT cells (*p* = 0.011, *p* < 0.001, and *p* < 0.001, respectively; Figure 4). There was no significant difference between the median values of *CV_M_* of the WT cells and the A634V cells (Figure 4).

Taking into account the notion that lower *CV_M_* values indicate cells closer to the state of tensional homeostasis (see Materials and Methods), our results demonstrate that E-cadherin variants compromising the juxtamembrane or the intracellular portions of the protein interfere with the cell’s ability to maintain tensional homeostasis. The higher values of the median *CV_M_* in the V832M and Mock cells may be partially explained by their lower values of the median 〈*M*〉, when compared with WT or the A634V cells (Figure 4 vs. Figure 3A) since, by definition, *CV_M_* and 〈*M*〉 are inversely related. However, 〈*M*〉 is not the sole determinant of *CV_M_* and the standard deviation, *SD_M_*, which is indicative of temporal variability of *M*(*t*), is also an important factor (recall that *CV_M_* = *SD_M_*/〈*M*〉). Indeed, we found that the WT cells had the lowest median values of *SD_M_*, while the R749W and V832M cells had the highest values, which is consistent with the differences in *CV_M_* between these cell types. A different interpretation of the above results follows from a consideration of temporal variability of traction forces at the FAs level.

In our previous study, we measured variability of individual traction forces applied to FAs in endothelial and vascular smooth muscle cells [21]. We found that their respective mean values of *CV_F_* did not change significantly, until their corresponding mean values of 〈*F*〉 reached a threshold value beyond which *CV_F_* precipitously decreased, indicative of FAs tensional homeostasis. In the present study, we did not find such a threshold of 〈*F*〉 in the AGS cell model. The mean values of *CV_F_* of the E-cadherin-transfected cells and Mock cells generally deceased with increasing mean 〈*F*〉 (Figure 5). However, the maximum mean values of 〈*F*〉 in the R749W (5.8 nN), V832M (5.5 nN), and Mock cells (4.3 nN) were smaller than that observed in the WT (8.0 nN) and A634V cells (6.8 nN) and hence, the corresponding values of *CV_F_* were higher in the R749W, V832M, and Mock cells than in the WT and A634V cells (Figure 5). This is in accordance with the observed lower values of *CV_M_* in the WT and A634V cells, than in the R749W, V832M, and Mock cells (Figure 4).

### 3.4. Collagen VI Enhances Traction Field Magnitude and Fluctuations

To study the effects of the ECM on cell mechanical response, we compared tensional homeostasis of cells seeded on gels micropatterned with Fn + Vt or with Col VI. We have previously shown that the Fn + Vt substrate was attractive for the A634V mutant and repulsive for the WT cells, whereas the Col VI substrate was attractive for the WT cells and repulsive for the A634V cells [22]. Herein, we verified that the differences observed for the WT, A634V mutant, and Mock cells were consistent between the Fn + Vt combination or the Col VI micropattern. However, the measurements carried out on the Col VI patterns yielded higher values of the median 〈*M*〉 in all cell types relative to the corresponding data obtained from the Fn + Vt experiments and this difference was significant in the Mock cells (*p* = 0.049, Figure 6A). The higher values of 〈*M*〉 may reflect higher numbers of FAs in the cells cultured on the Col VI micropatterns, when compared with that formed on the Fn + Vt micropatterns (Figure 6B), although the differences were not significant.

Median values of *CV_M_* obtain from the measurements on the Col VI micropatterns were higher in the WT and A634V mutant cells than the corresponding values obtained from the measurements on the Fn + Vt micropatterns (*p* = 0.009 and *p* = 0.236, Figure 6C), indicating that this ECM component induced increased traction fluctuation and is less favorable for tensional homeostasis.

Taken together, the results obtained on the different micropatterns suggest that the Fn + Vt combination decreases the variability of the traction field and supports a stable cell-ECM interplay, promoting tensional homeostasis.

### 3.5. Extracellular Mutant of E-Cadherin Does Not Affect Tensional Homeostasis in Cell Clusters

To further investigate whether the extracellular mutant of the E-cadherin protein affects tensional homeostasis, we carried out the traction microscopy measurements on clusters of WT and A634V cells. A salient assumption was that the A634V mutant might impair cell-cell force transmission which, in turn, might impact contractility and temporal variability of the traction field. WT clusters were composed of 2–14 cells and A634V clusters were composed of 2–10 cells. In both cases, we observed a significant increase in the median 〈*M*〉 in the clusters relative to single cells (*p* < 0.001 for WT cells; *p* = 0.024 for A634V cells, Figure 7A), possibly due to a significantly higher number of FAs in the clusters than in the single cells (*p* < 0.001 for both WT and A634V cells, Figure 7B). A mild effect, although not significant, was observed for median 〈*M*〉 of A634V clusters, when compared with the WT counterparts. We observed a non-significant decrease in the median *CV_M_* in the clusters relative to the single cells (Figure 7C). This marginal difference observed for traction variability between clusters and single cells suggests either that the extracellular mutant may not affect cell-cell force transmission, or if it does, then the intercellular force transmission has little effect on tensional homeostasis. Still, we are yet to provide definite evidence of the direct correlation between intercellular force transmission and either cell contractility or traction variability.

## 4. Discussion

Tensional homeostasis of malignant cells has been studied almost exclusively in the cases of breast cancer cells, in the context of mechanoreciprocity between the cell’s contractile forces and the stiffness of the extracellular matrix [4,5]. Those approaches describe tensional homeostasis as a static phenomenon. Here, we studied tensional homeostasis in AGS cells by focusing on the effect of E-cadherin and cancer-associated E-cadherin mutants on intracellular force transmission. Since cytoskeletal contractile forces vary over time, we approached tensional homeostasis as a dynamic process. Our strategy is consistent with the notion that homeostasis, in general, is continuously changing and oscillating around a set point. Moreover, the cellular environment is always ready to reset itself, but also to provide the reference point for a change if necessary for survival in an ever-changing environment [32].

Our findings demonstrate that in AGS cells, E-cadherin expression enhanced the magnitude and reduced temporal variability of their contractile moment. Importantly, the extent of these effects depended on the functional status of E-cadherin. The WT E-cadherin cells and the cells expressing the extracellular mutant exhibited high magnitude and low temporal fluctuations of the contractile moment, thus promoting tensional homeostasis. The cells with the juxtamembrane mutant exhibited nearly the same magnitude but a higher variability of the contractile moment in comparison with the WT cells. On the other hand, cells expressing the intracellular mutant variant exhibited both a lower magnitude and a higher variability of the contractile moment when compared with the WT cells. The higher variability of the contractile moment of the juxtamembrane and intracellular mutants interfere with the cell’s ability to maintain tensional homeostasis. In the absence of E-cadherin, as verified in the Mock cells, it was possible to detect the lowest magnitude and the highest variability of the contractile moment than the all E-cadherin-transfected cells, corroborating the hypothesis that E-cadherin expression is essential for tensional homeostasis in gastric epithelial cells.

The Fn + Vt micropattern substrates yielded lower magnitudes and lower variability of the contractile moment than the Col VI substrates in the E-cadherin-transfected cells, whereas in the Mock cells the Fn + Vt substrates yielded lower magnitude and higher variability of the contractile moment than the Col VI substrates. These findings suggest that the combination of Fn and Vt was more favorable for homeostasis than Col VI alone.

We may speculate that the WT cells and cells expressing the extracellular variant sustain an intact contractile actin cytoskeleton. Thus, the intracellular force transmission between E-cadherin and FAs across the cytoskeleton remains uninterrupted. Furthermore, this E-cadherin-FA crosstalk allows cells to develop a high level of cytoskeletal tension with relatively small temporal fluctuations, maintaining thereby tensional homeostasis [33]. Remarkably, it was documented that Myosin VI plays an important role in coupling of the E-cadherin juxtamembrane domain to the actin cytoskeleton [34]. It is also possible that Myosin VI may bind to E-cadherin along its full intracellular tail [34]. Thus, E-cadherin mutations affecting the juxtamembrane and the intracellular domains may induce a fragile interaction between E-cadherin and Myosin VI, consequently hindering intracellular force transmission among E-cadherin and FAs, and thus preventing maintenance of a high and stable cytoskeletal tension. In fact, a comprehensive characterization of a set of E-cadherin variants revealed that R749W and V832M impair the interaction with several molecular partners, namely p120, β-catenin, and PIPKIγ, compromising assembly of the cadherin-catenin complex, which mediates anchorage to the actin cytoskeleton and associated stability of cell–cell contacts [7,35]. As a result, there were deleterious effects in E-cadherin expression, localization and function, accompanied by alterations of cytoskeletal structures [7,36].

In an attempt to build up stable tension, cells may establish a high number of FAs and secrete ECM proteins to enrich the substrate, creating a favorable condition for tensional homeostasis. Accordingly, our results demonstrated that cells expressing the juxtamembrane or the intracellular E-cadherin variants exhibited an increased number of FAs, when compared to that detected in the WT cells or in the cells carrying the extracellular mutation. Importantly, it was previously reported that E-cadherin dysfunctional cells are able to produce and secret ECM components such as Laminin to survive and invade [37]. In contrast to cells expressing E-cadherin, the Mock cells could not develop high and stable cytoskeletal tension and thus they were the furthest from the state of tensional homeostasis.

It is noteworthy that our previous study on tensional homeostasis of AGS cells showed that cell clustering was much less effective for achieving homeostasis than in endothelial cells [20]. We found that temporal fluctuations of the contractile moment in clusters were slightly smaller than in single cells in both WT and Mock cells cultured on the Fn + Vt micropatterned gels. Together, these findings suggest that E-cadherin-mediated intercellular force transmission, which was present in WT clusters and absent in Mock ones, may have a mild impact on tensional homeostasis of AGS cells. Corroborating these data, our present work indicates that defective cell-cell adhesion, caused by E-cadherin extracellular mutations, might not strongly affect tensional homeostasis of AGS cells both at single cell and at cell cluster levels. Considering that the extracellular mutant is associated with gastric cancer, similar to the juxtamembrane and intracellular mutants, we postulate that this intriguing finding may underlie different penetrance of disease phenotypes—a major issue in research dedicated to hereditary diffuse gastric cancer.

Regarding the effect of ECM composition in our system, we verified that different matrix proteins did not result in qualitatively different behaviors of the cells tested. Based on the distinct adhesion affinities of the WT cells and the cells expressing the extracellular mutation towards the combination of Fn and Vt versus Col VI alone [22], we anticipated that the extracellular mutant cells would exhibit a higher value of 〈*M*〉 on the Fn + Vt micropatterns, than that observed on the Col VI micropatterns, whereas the WT cells would exhibit the opposite behavior. Surprisingly, our results showed that both WT cells and the extracellular mutants exhibited higher values of 〈*M*〉 and of *CV_M_* on the Col VI than on the Fn + Vt micropatterns, which is possibly related to differences in integrin types and/or levels. In this context, it has been reported that high traction forces are supported by α_5_β_1_ integrins, whereas less stable α_v_β_3_ integrins provide reinforcement of integrin-cytoskeleton linkages [38]. In the Mock cells, the difference observed between the median 〈*M*〉 on the Col VI and on the Fn + Vt micropatterns is further exacerbated. Taking into account that Mock cells display complete absence of E-cadherin, it appears that the presence of this protein (even if not functional) and the activation of its downstream signaling award cells an increased ability to adapt to distinct ECM compositions, as reflected in higher values of 〈*M*〉 and higher number of focal adhesions in cells transfected with WT or mutant forms of E-cadherin.

Ultimately, we would prefer to point out that in our previous publication, we were able to show a clear dependence of the magnitude of the traction field and the adhesion affinities of cells expressing WT and mutant E-cadherin cells towards the Fn + Vt combination and Col VI [22]. In that study, we used a different metric of the magnitude of the traction field—namely the sum of magnitudes of traction forces, which is different from the magnitude contractile moment that we have used in the present work. In fact, for the purpose of tensional homeostasis evaluation, we believe that it is more appropriate to use the magnitude of the contractile moment since it is directly associated to the mean cytoskeletal tension. Furthermore, the contractile moment accounts for the vectorial nature of traction forces and for the size of the cell, whereas the sum of the magnitudes of traction forces does not.

## 5. Conclusions

A breakdown of tensional homeostasis is a hallmark of epithelial cancers. Here, we showed that cancer-associated variants of E-cadherin located at the juxtamembrane or at the intracellular region of the protein might lead to loss of tensional homeostasis in AGS cells, in contrast to extracellular mutants. The behavior of the cells expressing an extracellular mutation was indeed similar to that of the WT cells in the sense that it was closer to the state of tensional homeostasis than the cells carrying juxtamembrane or intracellular mutations. Overall, our data suggest that juxtamembrane and the intracellular domains of E-cadherin are critical for tensional homeostasis by establishing an E-cadherin-cytoskeletal linkage, which sustains cellular tension. This work provides the first evidence that specific E-cadherin mutations are detrimental for tensional homeostasis, contributing to the disease progression.

## Figures and Tables

**Figure 1 cancers-14-02690-f001:**
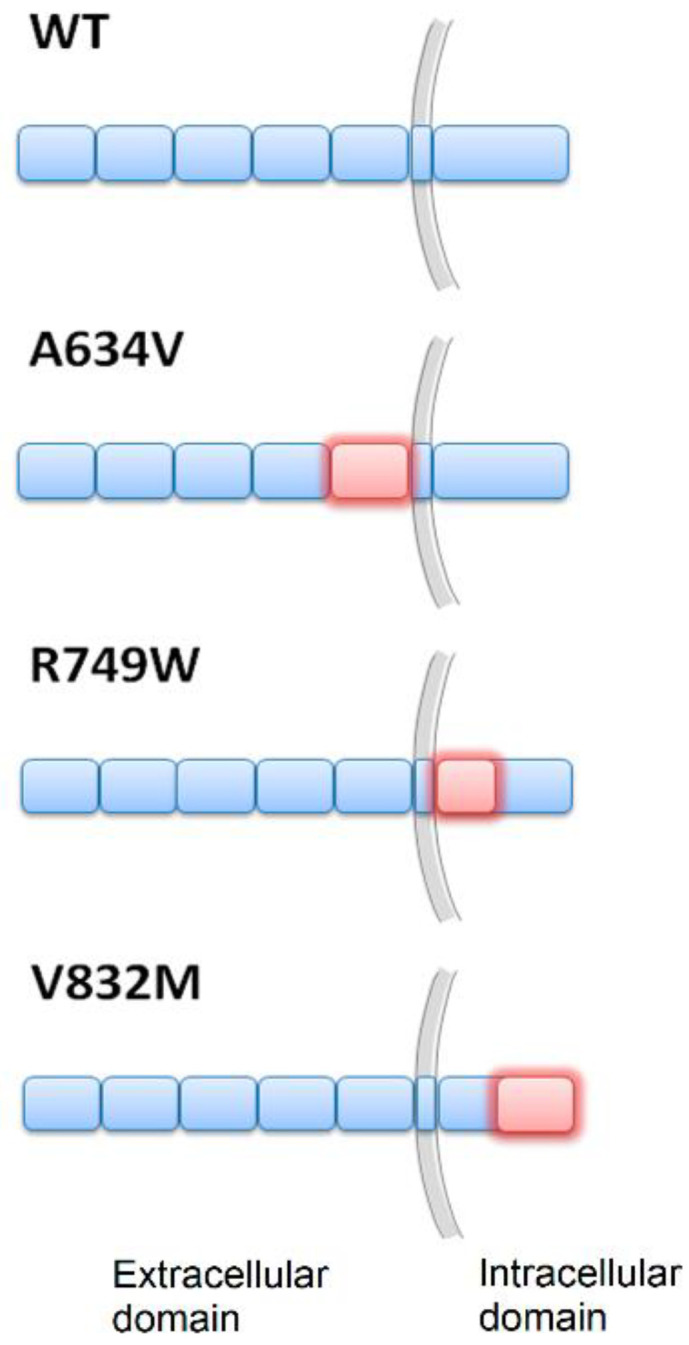
A schematic depiction of the localization of the different E-cadherin mutations (red) tested.

**Figure 2 cancers-14-02690-f002:**
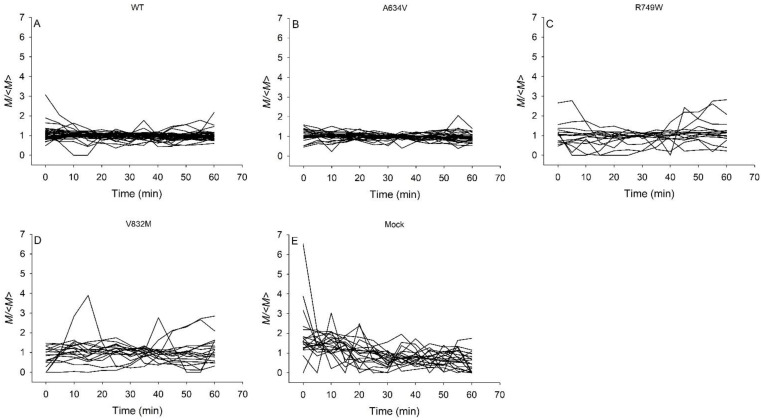
Impact of E-cadherin variants on temporal fluctuations of the contractile moment. Time lapses of normalized contractile moments of different types of AGS cells cultured on the combination of Fibronectin and Vitronectin micropatterns over the course of 60 min experiments. The graphs of WT cells (**A**) and A634V cells (**B**) exhibit smaller temporal fluctuations than the graphs of R749W cells (**C**), V832M cells (**D**), and Mock cells (**E**). Contractile moment (*M*) was normalized by its time-averaged value (〈*M*〉). Different lines correspond to different cells.

**Figure 3 cancers-14-02690-f003:**
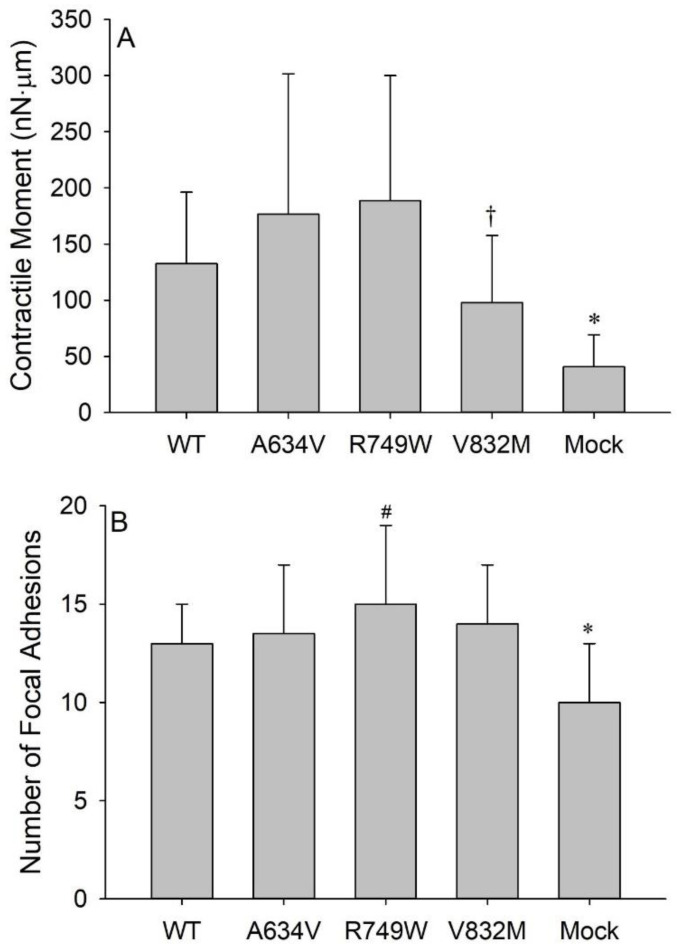
Impact of E-cadherin variants on the contractile moment and focal adhesion number. (**A**) Median values of time-averaged contractile moments of WT, A634V, R749W, V832M, and Mock AGS cells cultured on the combination of Fibronectin and Vitronectin micropatterns. The Mock cells exhibit a significantly smaller contractile moment than the other cell types (* *p* < 0.05), whereas the V832M cells exhibit a significantly smaller contractile moment than the A634V cells (^†^ *p* < 0.05). (**B**) Median values of the number of focal adhesions (FAs) of WT, A634V, V832M, R749W, and Mock AGS cells cultured on the combination of Fibronectin and Vitronectin micropatterns. The Mock cells exhibit a significantly smaller number of FAs than the other cell types (* *p* < 0.05), whereas the R749W cells exhibit a significantly greater number of FAs than the WT cells (^#^
*p* < 0.05). Graphs represent median ± MAD.

**Figure 4 cancers-14-02690-f004:**
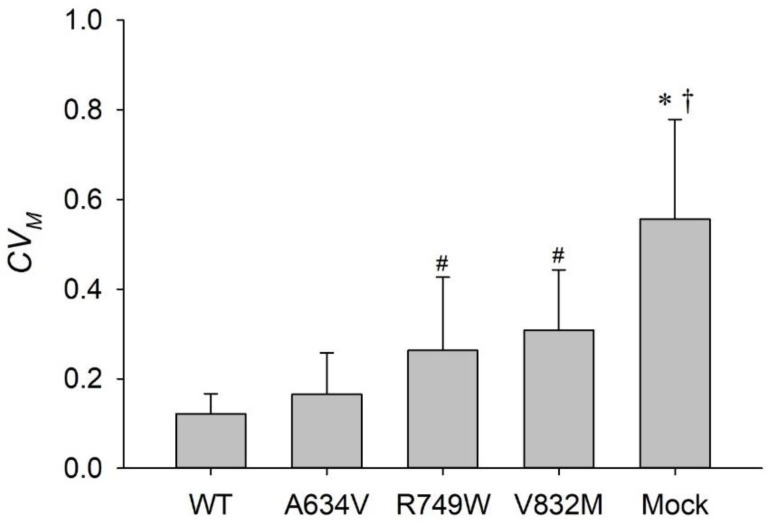
Impact of E-cadherin variants on tensional homeostasis. Median values of the coefficient of variation of the contractile moment (*CV_M_*) of WT, A634V, V832M, R749W, and Mock AGS cells cultured on the combination of Fibronectin and Vitronectin micropatterns. The Mock cells exhibit a significantly greater *CV_M_* than the WT, A634V, R749W (* *p* < 0.05), and V832M (^†^ *p* < 0.1), whereas the R749W and V832M exhibit a significantly greater *CV_M_* than the WT and A634V cells (^#^
*p* < 0.05). Graphs are median ± MAD.

**Figure 5 cancers-14-02690-f005:**
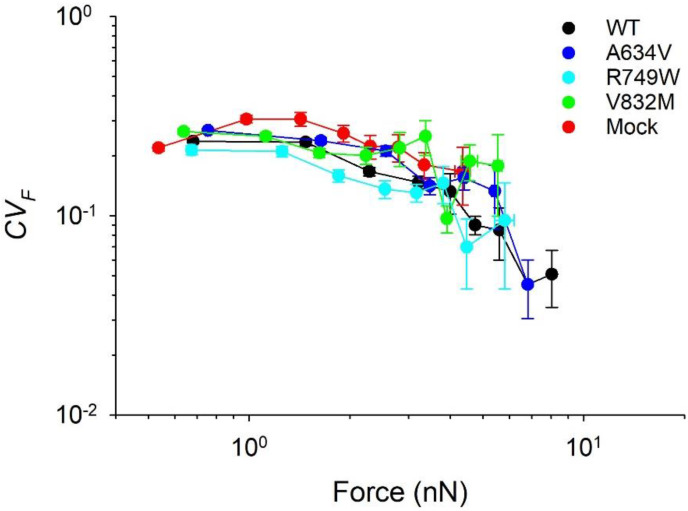
Impact of E-cadherin variants on tensional homeostasis of focal adhesions. With increasing of the time-averaged focal adhesions traction forces, their coefficient of variation (*CV_F_*) decreases. Different colors correspond to different cell types cultured on the combination of Fibronectin and Vitronectin micropatterns: WT cells (black), A634V cells (blue), R749W cells (cyan), V832M cells (green), and Mock cells (red). Data show mean ± standard error.

**Figure 6 cancers-14-02690-f006:**
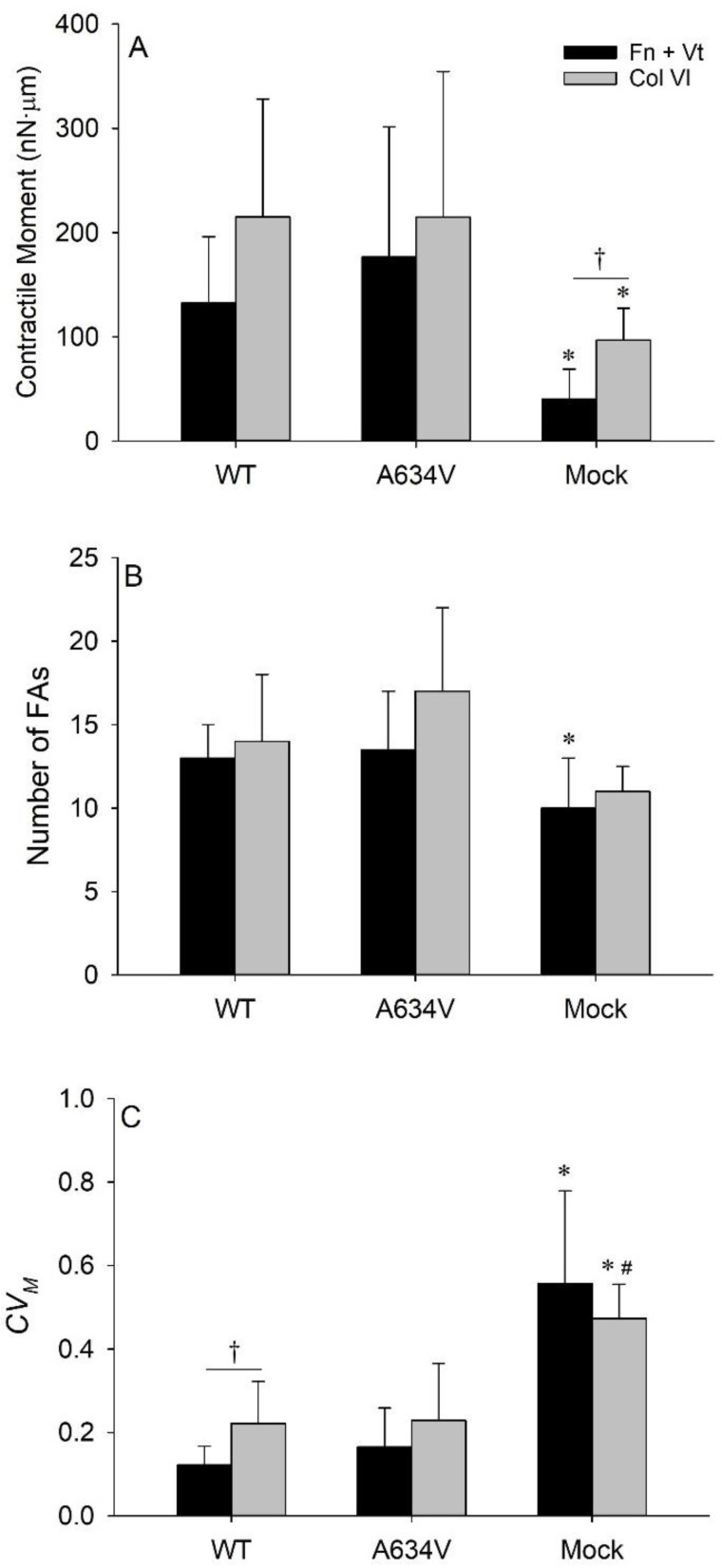
Effects of matrix proteins on the contractile moment and its temporal variability. Comparison of the median values of the time-averaged contractile moments (**A**); of the corresponding median numbers of focal adhesions (FAs) (**B**); and of the corresponding median values of the coefficient of variation of the contractile moment (*CV_M_*) (**C**) of the WT, A634V, and Mock cells cultured on the combination of Fibronectin and Vitronectin micropatterns (Fn + Vt, black bars) and on Collagen VI (Col VI, gray bars). The Mock cells exhibit significantly different values between the contractile moments obtained on the Fn + Vt and those obtained on Col VI micropatterns (^†^ *p* < 0.05). No significant differences in the median number of FAs between the two micropatterns is observed. Only the WT cells exhibit significantly different values of *CV_M_* obtained on the Fn + Vt versus Col VI micropatterns (^†^
*p* < 0.05). The graph bars represent median values ± MAD; * significantly different contractile moment and number of FAs (*p* < 0.05) than WT and A634V cells in the cases of Fn + VT and of Col VI micropatterns; * significantly different *CV_M_* (*p* < 0.05) than WT cells and ^#^ significantly different (*p* < 0.1) than A634V cells in the case of Col VI micropatterns.

**Figure 7 cancers-14-02690-f007:**
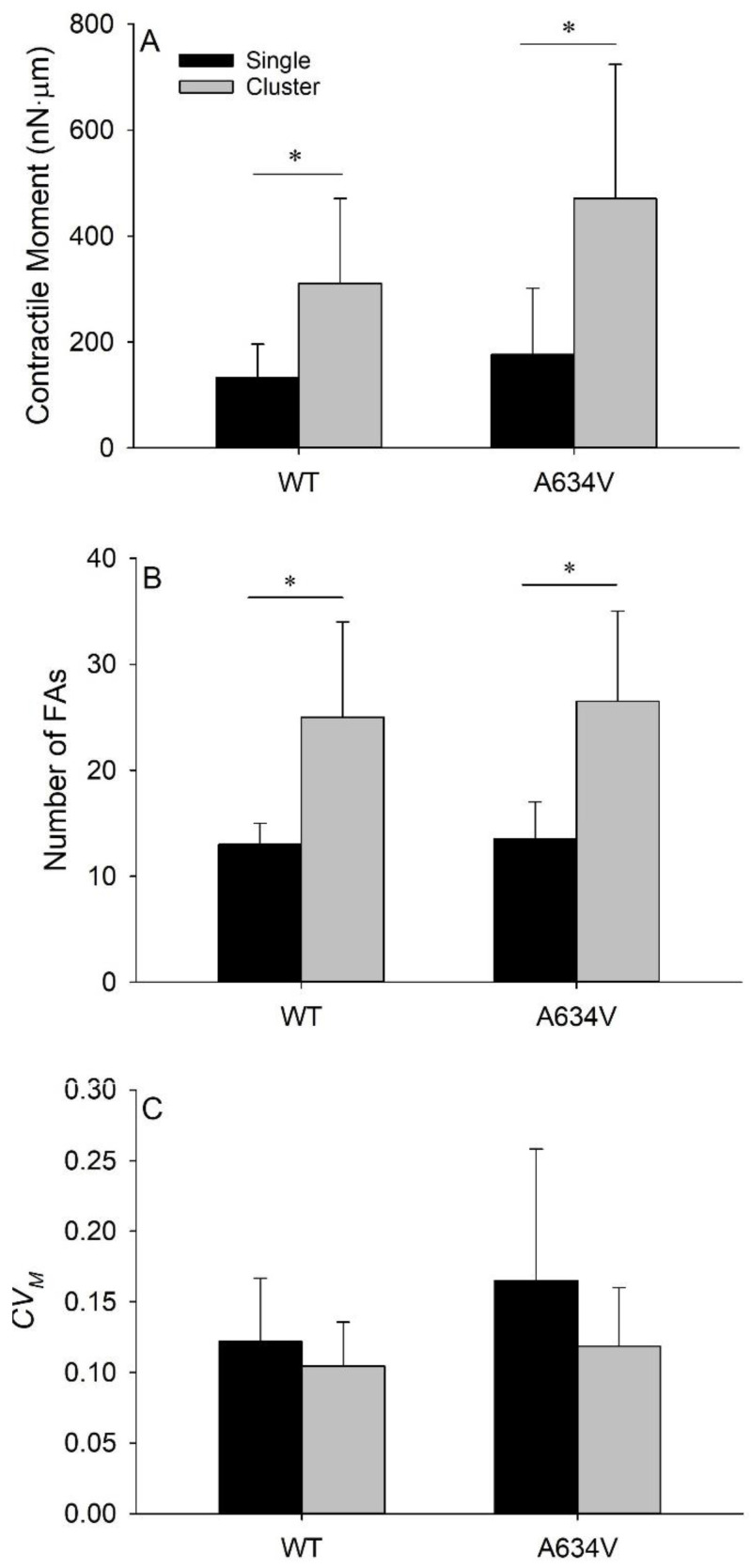
Tensional homeostasis in single cells and in cell clusters. Comparison of the median values of the time-averaged contractile moments (**A**); of the median numbers of focal adhesions (FAs) (**B**); and of the median values of the coefficient of variation of the contractile moment (*CV_M_*) (**C**) of single WT and A634V cells (black bars), and the corresponding clusters of cells (gray bars) cultured on the combination of Fibronectin and Vitronectin micropatterns. Clusters exhibit significantly higher values of the contractile moment and of the number of FAs than the single cells. No significant differences in the median values of *CV_M_* between single cells and clusters are observed. No significant differences in the median values of the contractile moment, the number of FAs, and *CV_M_* between WT single cells/cluster and A634V single cells/clusters are observed. The graph bars represent median values ± MAD; * significantly different contractile moment and number of FAs (*p* < 0.05).

## Data Availability

The datasets used and analyzed during the current study are available from the corresponding authors on reasonable request.

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
