# Peer review of "Differential Impacts on Tensional Homeostasis of Gastric Cancer Cells Due to Distinct Domain Variants of E-Cadherin"

_cancers, 2022, doi:10.3390/cancers14112690_

Round 1

Reviewer 1 Report

The study investigates the role of different domains of E-cadherin in the cellular contractile moments. Juxtamembrane and intracellular mutations infer increased instability in response to tensional forces. The effect is related to FA numbers. The results are of interest but the presentation is not optimal. For a lay person, it is hard to follow many analyses and their implications.

  1. In results, contractile moments is not explained and in methods the explanation is too technical. A more easily accessible explanation in the beginning of results would be appreciated.
  2. The location of the mutations is listed in methods but in results, a more extensive presentation of their structural and cell biological significance must be added. The reader wonders why effects of R749W, for example, might occur and how that relates to increased fluctuations.
  3. Fig 5, an inset with grey/black legend should be added. 
  4. Why were R749W and V832M analysed/presented? Seems like these should be included for completeness.

Author Response

We thank this reviewer for her/his useful comments and critique.  They helped us to improve the manuscript.  Our point-by-point response to specific comments are as follows.

  1. In results, contractile moment is not explained and in methods the explanation is too technical. A more easily accessible explanation in the beginning of results would be appreciated.

At the beginning of the results section (lines 172-180 of the revised manuscript), we provided an explanation which we believe would be more understandable to readers whose background is not engineering or physics. 

  1. The location of the mutations is listed in methods but in results a more extensive presentation of their structural and cell biological significance must be added. The reader wonders why effects of R749W, for example, might occur and how that relates to increased fluctuations.

In the results section, we added a new figure (Figure 1 of the revised manuscript) which shows a schematic depiction of the localization of the different E-cadherin mutations that were biomechanically tested in this study. 

Considering the remark regarding R749W, we have already discussed in the original submission potential molecular mechanisms associated with the connection between the E-cadherin and the contractile cytoskeleton which is facilitated by Myosin VI.  In the presence of R749V (and possibly V832M), this connection may become fragile, thus hindering force transmission between the contractile cytoskeleton and the E-cadherin and preventing thereby maintenance of high and stable (i.e., of low fluctuations) cytoskeletal tension (l. 474-486 of the revised manuscript).

  1. 5 inset with gray/black legend should be added.

Done.

  1. Why were R749W and V832M analyzed/presented. Seems like these should be included for completeness.

We are not sure what is the question.  Our guess is that the reviewer asked why R749W and V832M were not included in the analysis of the effect of matrix proteins on tensional homeostasis.  If so, the reason is that in our previous study (Ref. 22) we showed that the Fibronectin and Vitronectin combination is attractive for the A634V mutant cells and repulsive for the WT cells.  On the other hand, the Collagen VI is more attractive for the WT cells and repulsive for the A634V cells.  Thus, we selected these two cell types (as well as the Mock cells) and not the R749W and V832M cells, which do not have such contrasting affinities to these particular matrix proteins.  We explained this on l. 306-309 of the revised text.

Reviewer 2 Report

The paper entitled "E-cadherin variants affecting distinct protein domains impact

differently on tensional homeostasis of gastric cancer cells" mainly investigated the effect of E-cadherin mutations affecting distinct protein domains on tensional homeostasis of gastric cancer cells. The authors showed that mutations affecting distinct E-cadherin domains influenced different cell tensional homeostasis, and pinpointed the juxtamembrane and intracellular regions of E-cadherin as the key players in this process. Furthermore, Fibronectin and Vitronectin might modulate cancer cell behavior toward tensional homeostasis. The tension and adhesion of tumor cells are closely related to tumor invasion and metastasis. However, the following questions should be addressed to further improve the quality of this manuscript.

Major concerns:

  1. Figure 1 does not make it clear which cell lines of different colors represent. Please add it in figure legends. Further, it is suggested that while adopting the text description in Figure 4, graphic illustrations of cell lines represented by lines in different colors and shapes should be added next to the existing figure, so that readers can more intuitively understand the meanings of different lines in the figure.
  2. The authors mentioned in lines 204 and 205: that other factors may contribute to the differences in <M> observed across cell variants. No further discussion of the factors which may be associated with the medium values of <M> was found in the discussion section. What are the other factors? What impact will it have according to articles reported already?
  3. In line 233, the author mentioned that according to the results of Figure 2C vs Figure 2A, the higher values of the median CVM may partially be explained by their lower values of the median<M>, however, Figure 2C was not shown in this article. I strongly suggest the authors check it carefully.
  4. Figure 5 seems to be a bit crudely made. Although described in the text legend of figure 5, the grouping of Fn+Vt and Col was not identified in the figure, which is necessary and the figure may be difficult for readers to understand.
  5. The authors do not demonstrate why only A634V cells were selected for studying the effects of the ECM on cell mechanical response. What was the result of other variants mentioned above such as R749W (juxtamembrane domain mutant), and V832M (intracellular domain mutant)?
  6. The authors counted the contractile moment, number of FAs, and CVM of two different micropatterns in the three cells. How can these results explain the correlation between ECM and traction fluctuation and tensional homeostasis?
  7. This study wanted to show different E-cadherin variants affecting tensional homeostasis of gastric cancer cells. The experimental part, however, looks relatively straightforward and is almost entirely a demonstration of statistical results, which lacks direct image evidence. Microscopy images might be more intuitive.
  8. The authors found that mutations affecting distinct E-cadherin domains influenced different cell tensional homeostasis, and pinpointed the juxtamembrane and intracellular regions of E-cadherin as the key players in this process. However, it does not elaborate on the molecular mechanism through which the change of domain is affected. It is suggested to supplement the molecular mechanism of different mutants leading to the difference in tension homeostasis.

minor concerns:

  1. Please explain the criterion for the selection of the statistic threshold. For example,  p < 0.1 was used in statistics in figure 5 while in most cases the authors chose P<0.05 as the statistical threshold.
  2. There are some grammatical and lexical errors, such as ‘differently’ in line 31. The authors are advised to recheck the details and spelling of the manuscript.

Author Response

We appreciate the reviewer’s general remarks recognizing the significance of our study.  We thank the reviewer for her/his useful comments and critique.  They helped us to improve the manuscript.  Our point-by-point response to specific comments are as follows.

Response to Major Concerns:

  1. Figure 1 does not make it clear which cell lines of different colors represent. Please add it in figure legends. Further, it is suggested that while adopting the text description in Figure 4, graphic illustrations of cell lines represented by lines in different colors and shapes should be added next to the existing figure, so that readers can more intuitively understand the meanings of different lines in the figure.

We realize that different colors in Fig. 1 were confusing.  On each panel (A, B, C, D, E) different colors represented different cells of the same type, whereas the same color on different panels did not correspond to the same cell line.  In the revised text, Fig. 1 becomes Fig. 2 and all lines now had the same color (black).  We hope this clears the confusion.  Considering Fig. 4 (now Fig. 5 in the revised text), we added a figure legend explaining the meaning of different colors.

  1. The authors mentioned in lines 204 and 205: that other factors may contribute to the differences in <M> observed across cell variants. No further discussion of the factors which may be associated with the medium values of <M> was found in the discussion section. What are the other factors? What impact will it have according to articles reported already?

Other factors include the magnitude of traction forces acting at individual focal adhesions, extent of cell spreading, and cell shape, as indicated on l. 235-237 of the revised text.

  1. In line 233, the author mentioned that according to the results of Figure 2C vs Figure 2A, the higher values of the median CVM may partially be explained by their lower values of the median<M>, however, Figure 2C was not shown in this article. I strongly suggest the authors check it carefully.

Sorry for the typo.  It should have been Figure 3 vs. Figure 2A.  In the revised text, because of adding new Figure 1, Figure 3 is now Figure 4 and Figure 2A is now Figure 3A (l. 268-270 of the revised text).

  1. Figure 5 seems to be a bit crudely made. Although described in the text legend of figure 5, the grouping of Fn+Vt and Col was not identified in the figure, which is necessary and the figure may be difficult for readers to understand.

Done (now Figure 6).

  1. The authors do not demonstrate why only A634V cells were selected for studying the effects of the ECM on cell mechanical response. What was the result of other variants mentioned above such as R749W (juxtamembrane domain mutant), and V832M (intracellular domain mutant)?

In our previous study (Ref. 22) we showed that the Fibronectin and Vitronectin combination is attractive for the A634V mutant cells and repulsive for the WT cells.  On the other hand, the Collagen VI is more attractive for the WT cells and repulsive for the A634V cells.  Thus, we selected these two cell types (as well as the Mock cells) and not the R749W and V832M cells, which do not have such contrasting affinities towards these particular matrix proteins.  We explained this on l. 306-309 of the revised text.

  1. The authors counted the contractile moment, number of FAs, and CVM of two different micropatterns in the three cells. How can these results explain the correlation between ECM and traction fluctuation and tensional homeostasis?

We speculated that this might be related to different integrin types/levels expressed on two different ECM substrates (l. 519-521).  However, we do not have a good insight into potential biophysical and/or molecular mechanisms.

  1. This study wanted to show different E-cadherin variants affecting tensional homeostasis of gastric cancer cells. The experimental part, however, looks relatively straightforward and is almost entirely a demonstration of statistical results, which lacks direct image evidence. Microscopy images might be more intuitive.

We agree that microscopy images would help (although we would argue that they would be more intuitive).  However, the focus of this study was to provide a biomechanical aspect of tensional in gastric cancer cells.  Since tensional homeostasis is a biomechanical phenomenon (i.e., the ability of cells to maintain a consistent level of cytoskeletal tension with small temporal fluctuations around a set point), measurements of forces, moments, and their dynamics occupied a central place in this study.   With all its limitations, we believe that our biomechanical approach provides the first evidence that specific E-cadherin mutations are detrimental for tensional homeostasis, contributing to the disease progression.

  1. The authors found that mutations affecting distinct E-cadherin domains influenced different cell tensional homeostasis, and pinpointed the juxtamembrane and intracellular regions of E-cadherin as the key players in this process. However, it does not elaborate on the molecular mechanism through which the change of domain is affected. It is suggested to supplement the molecular mechanism of different mutants leading to the difference in tension homeostasis.

We agree that providing explanations of the observed biomechanical behaviors in terms of molecular mechanisms would complement our results.  However, as we pointed out in the response to the previous comment, the scope of this study was the biomechanical aspect of tensional homeostasis in gastric cancer cells.  Thus, elaborating in details on the molecular mechanisms for the observed mechanical responses was beyond the scope of this study.  Where possible, we briefly discussed potential molecular mechanisms that could explain some of the observed behaviors.  For example, the roles of Myosin VI, p120, β-catenin, and PIPKIγ, in the biomechanical responses of the cells with the juxtamembrane mutations and with the intracellular mutations (l. 474-486).  We also addressed a possible role of different types of integrins in explaining the biomechanical responses of cells cultured on different matrix substrates (l. 519-521).

Response to Minor Concerns:

  1. Please explain the criterion for the selection of the statistic threshold. For example, p < 0.1 was used in statistics in figure 5 while in most cases the authors chose P<0.05 as the statistical threshold.

Using several statistical thresholds in a single study is common.  P-values below 0.05 indicate statistically significant results and p-values below 0.1 indicates marginally significant results.

  1. There are some grammatical and lexical errors, such as ‘differently’ in line 31. The authors are advised to recheck the details and spelling of the manuscript.

Grammar and spelling errors are corrected and the title is modified.

Reviewer 3 Report

The authors analyzed the effects of wild-type E-cadherin and cancer-associated mutants of this protein on mechanical parameters of gastric carcinoma cells, in particular cellular tension. Forced expression of wild-type E-cadherin in E-cadherin-deficient gastric cancer cells led to enhanced cellular contractility and reduced temporal variation of contraction. The mutant E-cadherins showed similar effects on the magnitude of the contractile moment, but mutants of the juxtamembrane and intracellular domain of E-cadherin also reduced temporal fluctuations, which is interpreted as reduced tensional homeostasis. A mutant of the extracellular domain behaved largely similar to wild-type E-cadherin. All constructs increased the number of focal adhesions to slightly different degrees. Authors also discovered differences tension parameters depending on the extracelluar matrix that was deposited as a substrate for the cells.

These interesting findings shed new lights on E-cadherin function using a sophisticated methodology, however, this needs to be explained better for a broader readership of Cancers, and results should be linked to known function and biochemistry of E-cadherin and, if possible, the cancer mutants.

1 It is obvious that the authors have a long-standing expertise in the biophysical measurement the use in this paper which are clear to scientist in the field, but less so to a boader readership of a cancer journal. Therefore, basic principles of what is actually measured should be illustrated.

The authors should explain what information is extracted from the images and how traction forces and cell tension can be deduced from this information.  

The authors should describe how focal adhesions were defined and how they were quantified.

2 A scheme showing localization of the mutations on the E-cadherin protein would be helpful, as well as a statement whether these mutations have been previously characterized in terms of their cell-cell adhesion potential, association with intracellular b-catenin/a-catenin complex and the cytoskeleton. In that context, it is strange that the intracellular catenin link is not mentioned in the discussion because alterations in this connection might explain defects of juxtamembrane and cytoplasmic mutations. .

3 It appears that the extracellular domain mutant behaves much like wild-type E-cadherin in the measurements presented here although it is similarly associated to gastric cancer as the juxtamembrane and cytoplasmic mutants. This limits somewhat the relevance of these findings for hereditary cancer development, which needs to be discussed.   

Author Response

We appreciate the reviewer’s general remarks recognizing the significance of our study.  We thank the reviewer for her/his useful comments and critique.  They helped us to improve the manuscript.  Our point-by-point response to specific comments are as follows.

  1. It is obvious that the authors have a long-standing expertise in the biophysical measurement the use in this paper which are clear to scientist in the field, but less so to a broader readership of a cancer journal. Therefore, basic principles of what is actually measured should be illustrated.

The authors should explain what information is extracted from the images and how traction forces and cell tension can be deduced from this information.  

The authors should describe how focal adhesions were defined and how they were quantified.

We attempted to provide a less technical explanation for our micropattern traction microscopy that could be understandable for a broader readership.  For example l. 110-113 and l. 121-125 explain why focal adhesions (FAs) form only at the micropatterned dots.  Since traction forces are applied at FAs, and we can measure traction forces, the number of FAs is the same as the number of traction forces acting at a given instant (137-138).  

We also provided a description of the contractile moment using a non-technical language (l. 172-180).

  1. A scheme showing localization of the mutations on the E-cadherin protein would be helpful, as well as a statement whether these mutations have been previously characterized in terms of their cell-cell adhesion potential, association with intracellular b-catenin/a-catenin complex and the cytoskeleton. In that context, it is strange that the intracellular catenin link is not mentioned in the discussion because alterations in this connection might explain defects of juxtamembrane and cytoplasmic mutations. 

A schematic depiction showing localization of the E-cadherin mutants is included in the revised text as Figure 1 (p. 5).

A brief discussion of potential roles of the roles of p120, β-catenin, and PIPKIγ, in the biomechanical responses of the cells with the juxtamembrane mutations and with the intracellular mutations is added (l. 480-486).

  1. It appears that the extracellular domain mutant behaves much like wild-type E-cadherin in the measurements presented here although it is similarly associated to gastric cancer as the juxtamembrane and cytoplasmic mutants. This limits somewhat the relevance of these findings for hereditary cancer development, which needs to be discussed. 

The behavior of the extracellular domain mutant is intriguing.  To further investigate whether the extracellular mutant affects tensional homeostasis, we carried out the traction microscopy measurements on clusters of WT and A634V cells.  We found a non-significant decrease in the median CVM in the clusters relative to the single cells. This marginal difference observed for contractile moment variability between clusters and single cells suggests either that the extracellular mutant may not affect cell-cell force transmission, or if it does, then the intercellular force transmission has a little effect on tensional homeostasis.  Still, we are yet to provide a definite evidence of the direct correlation between intercellular force transmission and either cell contractility or traction variability (l. 360-432 and Figure 7).

Round 2

Reviewer 3 Report

Authors have responded adequately to the concerns. Although, it is still a difficult read for non-experts of the used methodology, the overall meaning and outcome of this study is now satisfactorily explained.